# Current State of Melanoma Therapy and Next Steps: Battling Therapeutic Resistance

**DOI:** 10.3390/cancers16081571

**Published:** 2024-04-19

**Authors:** Anna Fateeva, Kevinn Eddy, Suzie Chen

**Affiliations:** 1Susan Lehman Cullman Laboratory for Cancer Research, Rutgers University, Piscataway, NJ 08854, USA; anna.fateeva@rutgers.edu (A.F.); kevinneddy@gmail.com (K.E.); 2Graduate Program in Cellular and Molecular Pharmacology, Rutgers University, Piscataway, NJ 08854, USA; 3Rutgers Cancer Institute of New Jersey, New Brunswick, NJ 08901, USA; 4U.S. Department of Veterans Affairs, New Jersey Health Care System, East Orange, NJ 07018, USA

**Keywords:** melanoma, targeted therapy, immunotherapy, therapeutic resistance, riluzole, mGluR1

## Abstract

**Simple Summary:**

Skin cancer is the most common cancer type worldwide, and melanoma is its most aggressive and deadly form. In the past few decades, targeted therapy and immunotherapy have drastically increased survival chances for melanoma patients. However, not all patients respond to these therapies, and the majority of patients experience tumor relapse after initial response to the treatment. Continuous efforts are required to not only develop new therapeutic agents to treat this deadly disease but also to understand the underlying mechanisms leading to resistance of the tumors to treatment. In this review, we summarize data on available therapeutics for the treatment of melanoma and current knowledge on the mechanisms of resistance to melanoma therapies.

**Abstract:**

Melanoma is the most aggressive and deadly form of skin cancer due to its high propensity to metastasize to distant organs. Significant progress has been made in the last few decades in melanoma therapeutics, most notably in targeted therapy and immunotherapy. These approaches have greatly improved treatment response outcomes; however, they remain limited in their abilities to hinder disease progression due, in part, to the onset of acquired resistance. In parallel, intrinsic resistance to therapy remains an issue to be resolved. In this review, we summarize currently available therapeutic options for melanoma treatment and focus on possible mechanisms that drive therapeutic resistance. A better understanding of therapy resistance will provide improved rational strategies to overcome these obstacles.

## 1. Introduction to Melanoma

Skin cancer is the most prevalent type of cancer in the United States (US) and possibly in the world. The total number of cases may be higher than current estimates, as some types of skin cancer, such as basal cell and squamous cell skin cancers, are not required to be reported to cancer registries [1]. One study estimated over 5 million annual cases in the US alone in the last decade [2]. The most common types of skin cancer are basal cell carcinoma (BCC) and squamous cell carcinoma (SCC), followed by melanoma. Among the rarer types are Merkel cell cancer, cutaneous T-cell lymphoma (CTCL), Kaposi sarcoma, skin adnexal tumors and sarcomas. Even though BCC and SCC are the most prevalent types of skin cancer, they are usually easily curable and not metastatic. They originate in basal and squamous skin cells, respectively. Melanoma, on the contrary, is not nearly as common—it accounts for only 1% of all skin cancers diagnosed [3]; however, it causes the majority of deaths from skin cancer. In 2024, over 100,000 new cases of invasive melanoma are predicted in the US, with approximately 8000 melanoma-related deaths, which constitutes about 90% of deaths attributed to skin cancer [1]. Melanoma originates in the pigment-forming cells of the skin, melanocytes, which produce melanin. Melanin has a photoprotective function in the skin, guarding against UV-induced DNA damage. Melanoma is categorized into cutaneous and non-cutaneous. Cutaneous melanoma (CM) comprises neoplasms that arise on most skin surfaces of the body that can be further subcategorized into chronically sun-induced melanomas and non-chronically sun-induced melanomas. The four most common molecular subtypes identified for CM have mutations in *BRAF*, *NRAS* or *NF1* or a triple wild-type genotype for these genes. Non-cutaneous melanomas are rare and are classified into acral, mucosal and uveal subtypes. Acral melanoma (AM) often presents on palms, soles, fingers, toes and under the nails; mucosal melanoma (MM) occurs in mucosal membranes, such as in the mouth and nasal cavities; and uveal melanoma (UM) originates in the ocular region. These rare melanoma types are very aggressive because they are difficult to detect and therefore are not diagnosed at early stages of tumor development [4].

There are several risk factors for developing skin cancer, which include chronic exposure to UV light, having a high number and large size of moles on the skin, as well as fair skin and light hair color. Additionally, familial history of skin cancer increases the chances of developing the disease in some cases due to the association with genetic mutations, including *CDKN2A*, *CDK4*, *MC1R* and mutations in the nucleotide excision repair machinery pathway [5,6].

The most common melanoma type is cutaneous melanoma. *BRAF* mutations constitute the most common genetic mutation present in about 50% of melanoma patients, but mutated *BRAF* is also detected in melanocytes [7,8]. Melanomas with mutated *BRAF* are usually located on parts of the body not typically exposed to the sun [9]. Over 90% of *BRAF* mutations occur at codon 600: the most common one is the substitution of glutamic acid for valine (V600E) [7]; other minor mutations include substitution of glutamic acid for lysine (V600K), aspartic acid (V600D) and arginine (V600R) [10]. BRAF is a participant in the MAPK signaling cascade downstream of RAS (Figure 1). As such, most mutations in BRAF lead to constitutive activation of this pathway, triggering aberrant cell proliferation and inhibition of apoptosis. Both ultimately lead to tumor development and progression with additional mutations acquired, such as PTEN [11,12]. Mutations in MEK, the protein downstream of BRAF in the MAPK signaling cascade, occur less frequently than in BRAF, with 6–7% incidence that is mostly in MEK1. Upstream of BRAF, NRAS has a mutational frequency of 15–28% [7,13]. Identification of these hot-spot mutations led to the development of targeted-therapy inhibitors (Figure 1).

## 2. Treatment Options Available for Cutaneous Melanoma and Their Efficacy

### 2.1. Radiation Therapy, Topical Therapeutics and Chemotherapy

Surgical tumor removal has been the standard of care for patients with primary melanoma. Radiation therapy, which is common for many other cancer types, has not gained widespread use in melanoma, as skin tumors are usually radioresistant. Instead, therapeutic agent administration is a more likely treatment option for most patients. However, radiotherapy remains an option for patients with inoperable tumors, as well as imiquimod cream, a local immunomodulator prescribed to some patients with early-stage melanoma [14,15].

Treatment with dacarbazine, a chemotherapeutic agent that was introduced in the 1970s, was the standard of care for melanoma patients until targeted therapy was introduced in 2011 (Table 1). Dacarbazine alkylates DNA non-specifically to block DNA replication [16]. The rate of objective tumor responses in patients on dacarbazine ranged from 13 to 20%, with nearly all responses being partial [17]. In addition, dacarbazine caused severe adverse effects (AE) in patients [16].

### 2.2. Targeted Therapy

BRAF is the most commonly mutated gene in melanoma, and it is therefore an attractive candidate for targeted therapy. One of the first inhibitors developed against mutated BRAF was sorafenib, which is a multikinase inhibitor that targets CRAF, both wild-type and mutant BRAF, and multiple receptor tyrosine kinases (RTKs) [43]. However, the efficacy of sorafenib was limited both as a single agent and in combination with chemotherapeutics, likely due to its weak affinity for BRAF [44,45]. As a result, inhibitors that could bind specifically to mutated BRAF were developed, such as vemurafenib, dabrafenib and encorafenib (Table 1). These inhibitors bind to the ATP-binding pocket of BRAF with increased affinity for BRAF V600E mutation which enhances their selectivity [24,46,47]. They demonstrated increased efficacy over chemotherapeutic agents with a better dose-dependent tumor inhibition in preclinical studies, a higher rate of objective responses (OR) and improved overall survival (OS) in clinical trials [25,26,47,48,49,50]. Despite the improved efficacy of BRAF inhibitors (BRAFi) over chemotherapy, in most cases disease progression occurs after 6–7 months of treatment due to acquired resistance [25,51]. The molecular mechanism underlying this induced resistance is often the reactivation of MAPK signaling through initiation of the MEK cascade [43]; this will be discussed in detail later in the review. The first MEK1/2 inhibitor (MEKi)—trametinib—was developed along with BRAF inhibitors and was shown to also have improved treatment outcomes over chemotherapeutic agents. Unfortunately, patients also developed resistance to the inhibitor shortly after the beginning of treatment [51,52]. Therefore, reactivation of the MAPK cascade after treatment with BRAFi led to the use of a combination of BRAFi and MEKi to overcome therapy resistance and improve treatment outcomes. Three different combinations of BRAFi and MEKi were approved for melanoma patients: dabrafenib/trametinib, vemurafenib/cobimetinib and encorafenib/binimetinib [43]. These approaches increased overall response and survival rates for patients with advanced melanoma to 50–70% [53,54,55,56,57,58,59,60]. The combination therapy, both as a standalone therapy and as an adjunctive therapy, demonstrated strong efficacy in patients [35]. Although effective, BRAFi/MEKi combinations had their shortcomings, mostly in the form of adverse effects, which ranged from mild to severe and occurred in 45–70% of patients who participated in the COMBI-D, COMBI-V, coBRIM and COLUMBUS clinical trials [61].

### 2.3. Immunotherapy

The development of immunotherapeutic agents, such as inhibitors of cytotoxic T-lymphocyte-associated protein 4 (CTLA-4) and programmed cell death protein 1 (PD-1), was of immense significance, as their modes of action are independent of the genetic profiles of any cancers, including melanoma. One of the first immunotherapy options available for the treatment of melanoma was interferon alpha (IFNα) as an adjuvant treatment (Table 1). It modulates the tumor immune environment by increasing the number of tumor-infiltrating cells, decreasing circulating T-regulatory cells, altering cytokine levels and activating transcription 1 (STAT1) and transcription 3 (STAT3) signal transducers [20]. The OS rate was shown to improve in patients treated with IFNα; however, more effective therapies have since replaced IFNα as a melanoma treatment option. Another early immunotherapy option was a high-dose interleukin-2 (IL-2) regimen that modulated immune signaling (Table 1). With response rates (RR) at about 16%, IL-2 therapy did not differ much in efficacy from chemotherapy; however, it offered an alternative to patients who were not responding to dacarbazine, and the response seemed to be more durable [22]. The first immune checkpoint inhibitor introduced was ipilimumab, an antibody against the CTLA-4 receptor present on the surface of T-cells. It was approved for use in melanoma patients in 2011 (Table 1) [27]. CTLA-4 is a co-receptor on the surface of T-cells that is responsible for their activation. When blocked by CTLA-4 specific antibodies, immune responses against tumor cells were observed through an increase in various immune cells (tumor-specific T-cells and non-tumor-specific immune cells) that promoted an increase in anti-tumor activity but also toxicity [62]. Ipilimumab was shown to be effective in patients previously treated with dacarbazine or IL-2, with a median OS of 10.1 months, an overall response rate of 10.9% and an objective response observed in 60.0% of patients that was maintained for at least 2 years in a phase III clinical trial. Grade 3 or 4 adverse effects that could be life-threatening and/or treatment-limiting were reported to occur in 10–15% of patients [28]. Another study, using a combination of ipilimumab and dacarbazine in previously untreated metastatic melanoma patients, confirmed the effectiveness of ipilimumab over the traditional chemotherapy regimen. The OS rates on the combinational ipilimumab/dacarbazine treatment were 11.2 months vs. 9.1 months on dacarbazine alone. However, the rate of adverse effects was increased in the combinational therapy group to 56.3% compared to 27.5% in a single-treatment arm using dacarbazine or placebo [63].

Another promising immunotherapy axis is programmed cell death protein 1 (PD-1)/programmed death-ligand 1 (PD-L1) [27]. PD-1 is a receptor on T-cells that is responsible for maintaining self-tolerance and the protection of host cells against autoimmunity by suppressing the activity of host T-cells. PD-L1 is the endogenous ligand for PD-1 that is expressed in many normal and immune cells. Binding of PD-L1 to PD-1 activates the downstream immune signaling cascade that deactivates T-cells. Overexpression of PD-L1 has been found in several cancer types and is thought to promote evasion from immune detection in tumor cells [64,65,66]. Antibodies specific to PD-1 and PD-L1, respectively, were found to inhibit solid tumor growth in several studies [67,68] and in clinical trials with melanoma patients [29,31,69,70,71]. Two PD-1 antibodies, pembrolizumab and nivolumab, were approved by the FDA for melanoma patients in 2014 (Table 1). Pembrolizumab proved to be more effective than ipilimumab, with an estimated 6-month progression-free-survival (PFS) rate of 46.5% compared to that of 26.5% for ipilimumab. The response rate (RR) was also significantly improved for pembrolizumab, with 32.5% compared to an RR of 11.9% for ipilimumab. Additionally, the incidence of adverse effects was also lower for pembrolizumab (10–13%) than for ipilimumab (19.9%) [31]. Nivolumab demonstrated effectiveness in patients with ipilimumab-refractory tumors in a phase III study when compared to a chemotherapy treatment, with objective responses of 32% vs. 17% [30]. Another phase III study that involved patients with previously untreated melanoma tumors showed an overall survival of 72.9% after 1 year on nivolumab versus 42.1% after 1 year on dacarbazine. The objective response rate (ORR) was 40.0% for nivolumab versus 13.9% for dacarbazine, establishing survival benefits with nivolumab over dacarbazine. Grade 3 or 4 adverse effects rates were also slightly lower in patients on nivolumab (11.7%) than in those on dacarbazine (17.6%) [29]. Thus, immunotherapy approaches paved the way to a new era of treatment for melanoma patients.

Given the success in preclinical models testing combinations of immune checkpoint inhibitors in the treatment of melanoma, clinical trials were held to assess the benefits of using such combinations in patients [34,72]. The combination of nivolumab and ipilimumab proved to be a successful therapeutic strategy in a phase III clinical trial, with an improved median OS rate in comparison with nivolumab monotherapy. This approach was approved by the FDA in 2015 (Table 1) [34]. However, despite showing improved immediate efficacy of the combination, including increased responses, the long-term survival data demonstrated a 6.5-year OS in 50% of participants in the CheckMate 067 phase III trial [34]. Unfortunately, there was a higher risk of severe adverse effects in patients on the combination compared to those on the monotherapy. This sparked an interest in identifying prognostic biomarkers that could single out patients who are likely to benefit from immunotherapy treatment [73].

Antibodies to block PD-L1 were also explored. An in vitro study revealed a higher potency of PD-L1 compared to PD-1 antibodies based on a T-cell reporter platform using EC_50_ values as an indicator [74]. To this end, atezolizumab, a PD-L1 antibody, was tested in a phase III clinical trial in combination with BRAFi or MEKi, or with vemurafenib or cobimetinib, respectively. The overall survival rates between the combination group and the control group (which received a combination of vemurafenib and cobimetinib as standard of care at the time) were similar. However, PFS rates for the combination group were significantly better than those for the control group at rates of 21.0 months vs. 12.6 months, suggesting a prolonged duration of efficacy when using the PD-L1 antibody atezolizumab with either vemurafenib or cobimetinib. Therefore, it was established that the combination of PD-1/PD-L1 inhibitors with BRAF and MEK inhibitors demonstrated a similar response rate to the combination of PD-1 blockers but that it provided longer-lasting benefit [75,76,77,78,79]. In 2020, the FDA approved the use of atezolizumab in combination with cobimetinib and vemurafenib for patients with advanced unresectable melanoma harboring the BRAF V600E mutation (Table 1) [37].

In 2022, the FDA approved the first lymphocyte-activation gene 3 (LAG-3) immune checkpoint inhibitor, relatlimab, in combination with nivolumab for the treatment of advanced melanoma (Table 1). LAG-3 is a receptor on the surface of T-cells that regulates the activity of T-cell mediated cell proliferation and function. The combination of relatlimab and nivolumab was tested in patients with resectable and previously untreated/unresectable melanoma [39,80]. When used in the neoadjuvant setting, the 1- and 2-year recurrence-free survival rates for patients with a pathological response were 100% and 92%, respectively, compared to 88% and 55% for patients without a pathologic response. The absence of grade 3 or 4 adverse effects was also noted [80]. The trial involved patients with either previously untreated or unresectable melanoma tumors who demonstrated progression-free-survival rates of 47.7% at the 1-year timepoint with relatimab/nivolumab versus 36.0% for nivolumab monotherapy. Grade 3 or 4 adverse effects occurred in 18.9% of patients in the combination therapy group compared to 9.7% of patients in the monotherapy group [39]. As seen in these trials, LAG-3 inhibitor was unable to reduce therapy toxicity; however, it provides an alternative treatment for patients with refractory tumors [40].

A therapeutic option of a different class is talimogene laherparepevec (T-VEC), an oncolytic virus that is used for the treatment of unresectable and advanced melanoma. It was approved by the FDA in 2015 (Table 1). T-VEC induces the expression of cytokine granulocyte–macrophage colony-stimulating factor (GM-CSF) that recruits and activates antigen-presenting cells that mobilize the host’s immune system to the tumor [81,82]. T-VEC was demonstrated to increase survival rates by approximately 4 months when compared to GM-CSF monotherapy. It was proposed for use as a component of combinational therapy with other therapeutic agents based on its mechanism of action [81,82,83,84].

In 2024, the first cell therapy approach was approved by the FDA for the treatment of melanoma patients with unresectable or metastatic melanoma that has been previously treated with anti-PD-1 and BRAFi [42]. Lifileucel is a tumor-derived autologous T-cell immunotherapy based on in vitro T-cell expansion. This line of treatment demonstrated OR rates of 31.5% in a phase II clinical trial [85]. A phase III clinical trial is currently underway, with results expected in 2028–2030 (NCT05727904).

Another direction that has potential in the treatment of melanoma is photodynamic therapy (PDT). PDT has had some success in treating several cancers, such as head and neck, breast, lung, non-melanoma skin cancer and esophageal cancer [86,87,88,89,90,91]. It typically involves using photosensitizer molecules that target the diseased tissue. Lasers with certain light wavelengths activate photosensitizers to yield singlet oxygen, which is oxygen in an excited state, that will act by the same principle as reactive oxygen species (ROS) to cause further damage to cancer cells. Such damage includes, but is not limited to, increased immune responses and DNA damage, which lead to cell death. Through its mechanism of action, PDT limits damage to the normal tissue and demonstrates increased selectivity and low AEs [92,93,94]. In melanoma specifically, PDT has been shown to cause apoptosis, tumor vasculature damage and anti-tumor immune responses [92]. Additionally, the combination of PDT with immunotherapy is a promising axis with favorable results demonstrated in vivo and in situ [92,95,96,97]. Approaches combining PDT with nanoparticles and phytochemical products are also being explored for their efficacy against melanoma [98,99]. PDT has not yet gained widespread application, in part due to its relatively low efficiency. However, it is a promising direction in the field of melanoma therapy.

## 3. Available Treatment for Acral, Mucosal and Uveal Melanoma

Acral melanoma (AM) and mucosal melanoma (MM) have been noted to be biologically and clinically distinct from cutaneous melanoma (CM) [4]. Low incidence rates of these melanoma subtypes have made it very difficult to determine the effectiveness of treatment due to problems with patient recruitment and sample size. AM constitutes 2–3% of melanoma cases, and MM accounts for approximately 1.2% of all melanoma cases [100,101]. The rarity of AM and MM greatly hinders the identification of critical genes involved in the onset and progression of AM and MM, impeding the development of appropriate treatment options. Because of that, retrospective studies and subgroup analysis of clinical trials not specific to patients with AM or MM subtypes of melanoma had to be conducted to obtain data on available therapeutic options for these patients.

For AM, current standard of care involves surgical removal of the tumor [4]. Given the specificity of this subtype, depending on the location of the tumor, plastic surgery is often required, and amputation could be an additional treatment option in the case of subungual melanomas. Some retrospective studies have demonstrated that a non-amputation treatment option coupled with a full-thickness skin graft is an effective alternative to amputation [102,103]. However, at this time, amputation remains the recommended strategy for subungual melanoma [4]. Interferon alpha has been used as an adjuvant therapy for AM patients with a high risk of relapse and it was demonstrated to reduce the risk of melanoma recurrence; however, its efficacy remains controversial, with a lack of consistency in overall survival improvement for AM patients [104,105,106].

Activating mutations in BRAF are detected only in 15–20% of AM and 3–11% of MM tumors, rendering BRAF/MEK inhibitors effective in only a small population of patients [52,107]. In a retrospective analysis of a clinical trial involving BRAF inhibitors, the ORRs were 38.1% for AM patients and 20% for MM patients [108]. Another study suggested that these inhibitors are just as effective for AM and MM patients with a BRAF V600 mutation as for CM patients with the same mutation [109]. Activating mutations in the receptor tyrosine kinase KIT are more common than those in BRAF and have incidence rates of 36% for AM and 39% for MM patients [110]. Several clinical trials using imatinib, the most well-studied KIT inhibitor, were conducted and the responses to the inhibitor were found to be variable [111,112,113]. Even though some patients benefited from the KIT inhibitor therapy, in most cases the disease ultimately progressed within 1–2 years [4]. Interestingly, KIT was found to have ambiguous functions in cancer. There are multiple indications suggesting that KIT has tumor-suppressing functions in melanoma. Its low expression was associated with higher metastatic potential, and its expression has also been lost in the majority of melanoma cell lines [114,115,116,117]. On the other hand, in other cancer types such as acute myeloid leukemia, gastrointestinal stromal tumors and small-cell lung carcinomas, KIT was associated with tumor progression, with activating mutations in 70–80% of cases. Unlike BRAF and NRAS, KIT mutations were reported not to occur at the same codon in the majority of cases. Rather, they might be spread more heterogeneously throughout the gene, making the development of inhibitors for mutated KIT a challenge [117,118]. This heterogeneity could account for a lack of response to a specific KIT inhibitor in some patients [117]. Other KIT inhibitors, such as nilotinib, were explored for their efficacy in AM and MM, but they ultimately showed similar outcomes to imatinib [119,120].

Surgical excision of the tumor remains standard of care for MM patients. However, given the anatomic sites of the malignancy, morbidity rates for such procedures are higher for MM patients than those for AM and CM. Additionally, neoadjuvant therapy is gaining importance and wider clinical application for MM patients; however, to date, there is no systematic data on its use in clinical settings [4].

Currently, immunotherapy is the most conventional option for adjuvant therapy in AM and MM patients. However, due to the low incidence rates of these melanoma subtypes, data on the safety and efficacy of such immunotherapy is very limited. In a study comparing efficacy between nivolumab and ipilimumab for AM and MM patients, at the 4-year follow-up there was no significant improvement with nivolumab versus ipilimumab treatment; this is likely due to the low number of patient participants [121]. Retrospective analyses of several studies revealed that anti-PD-1 treatment is more effective than a high-dose of interferon α-2b in the adjuvant setting in CM patients but not in AM patients, which could be due to a lower tumor mutational burden in AM patients than in CM patients [122,123]. Despite the lack of sufficient data on the efficacy of immunotherapy in AM and MM, patients with a high risk of relapse will be considered for immunotherapy in an adjuvant setting. In the case of aggressive metastatic AM, pembrolizumab and nivolumab were shown to be effective for disease management to a degree similar to that in CM patients [124]. Several other studies reported conflicting outcomes on the efficacy of anti-PD-1 therapy in AM patients, with shorter OS and ORR rates observed compared to non-AM patients [125,126]. A study using a combination of nivolumab and ipilimumab demonstrated an improved ORR comparable with non-AM patients [127]. Retrospective analyses of several trials testing the efficacy of anti-PD-1 in advanced MM patients revealed lower responses, which is in line with other studies [124,126,128,129,130]. Similar to what was seen in CM, increased efficacy in combinational immune therapies over single therapy regimens was demonstrated in AM and MM, albeit to lower degrees than for CM patients [129].

Uveal melanoma (UM) is a form of ocular cancer. There are approximately 5 cases per 1 million people in the US registered annually [131,132], but the distal metastasis rate in UM is approximately 50% [133]. The management of UM is highly individualized. If the disease is localized, then laser therapy, radiation therapy and surgery are the primary treatment options depending on the size of the tumor and other accompanying circumstances. Chemotherapy is surprisingly ineffective for uveal melanoma, with a response rate of less than 1% [134]. However, in 2023, the FDA approved the use of the chemotherapeutic agent melphalan for UM patients with liver metastasis using a special hepatic delivery system which limits systemic toxicity of the drug (Table 1) [41]. A targeted approach to treating UM is hindered by the genetic distinction of UM from CM, and the rarity of UM greatly impedes the development of novel therapeutic options. Most clinical trial outcomes available for UM are retrospective analyses of larger trials. So far, no significant responses in immunotherapy or targeted-therapy clinical trials have been reported for metastatic uveal melanoma patients. Liver is the most common site of metastasis for UM. Apart from melphalan, direct intervention by means of surgery, hepatic artery embolization and radiofrequency ablation can be used [134]. A recent breakthrough is the development of tebentafusp, a T-cell receptor-bispecific molecule that targets both glycoprotein 100 and CD3. It was approved by the FDA in 2022 for the treatment of unresectable or metastatic UM in patients who are positive for HLA-A*02:01, which is the most prevalent and polymorphic major histocompatibility complex (MHC)-allele family in humans (Table 1) [38,135]. A recently concluded phase III trial found that tebentafusp treatment was effective for UM patients, with a 3-year OS of 22% in the treatment group compared to 17% in the control group [136]. Carvajal et al. described in detail the current state of clinical management of UM and potential therapeutic strategies for UM patients [135]. More studies are also available on UM, and this review is not an exhaustive source of information on the subject.

Despite the significant progress made in the past two decades much still needs to be done to advance available therapeutic options for melanoma patients, especially those with rare types of melanoma. As demonstrated in some cases with immune checkpoint inhibitors, targeted agents not specific to the tumor (sub)type can be successful and in fact preferential over targeted therapy to achieve better results in a variety of melanoma types. Therefore, it is of great importance to investigate and develop ‘multifaceted’ therapeutics capable of treating various subtypes of the disease.

## 4. mGluR1-Driven Melanoma

### 4.1. Glutamate in Cancer Development and Progression

Glutamate is a non-essential amino acid that has been largely implicated in both the central nervous system (CNS) and peripheral nervous system as a major excitatory neurotransmitter and a key regulator in learning and memory. Its link to neuropathology has been firmly established. However, glutamate is also an evolutionarily conserved signaling molecule found in various other non-neuronal tissues of the body that contributes to metabolism and structure, as well as to numerous signaling pathways [78,79,137,138,139]. Therefore, it is not surprising that aberrant glutamatergic signaling can lead to the pathogenesis of other diseases such as cancer. The main characteristic of cancer cells is their ability to proliferate uncontrollably. To sustain such intense expansion, tumor cells rewire metabolic pathways, including glutamate metabolism, to meet the increased demand by amplifying the supply of energy and nutrient material necessary for the synthesis of new biomolecules for dividing cells [140]. Glutamate is linked to the production of the major cellular energy source, adenosine triphosphate (ATP), through the tricarboxylic acid cycle (TCA). In cells, glutamate is produced from glutamine by the enzyme glutaminase (GLS) in a hydrolytic reaction with the formation of ammonia that takes place in the mitochondria. It is then converted to α-ketoglutarate by glutamate dehydrogenase; subsequently, α-ketoglutarate enters the TCA cycle to contribute to ATP production [141].

Cancer cells proliferate rapidly, causing a constant need for energy to support their viability and expansion. However, oxygen supply inside a tumor is often limited, leading to hypoxia and hindering the use of aerobic oxidative phosphorylation, which is the main energy source for non-neoplastic cells. Cancer cells then switch over to using anaerobic glycolysis as their major energy source, a phenomenon also known as the Warburg effect [142,143]. It has been reported for many different cancer types that tumors rewire cells to use glutamine/glutamate as the major metabolite, serving as an alternative energy source. This is known as ‘glutamine addiction’ [144,145]. One of the proposed mechanisms for such rewiring is the upregulation of GLS to increase available glutamate levels inside the cell [146,147,148,149].

Taken together, the notion of targeting glutamine/glutamate metabolism is a promising therapeutic strategy for cancer therapy. Additionally, glutamate is a ligand for multiple cell surface receptors that initiate downstream signaling cascades, including metabotropic glutamate receptors (mGluRs) and ionotropic glutamate receptors (iGluR). mGluRs have been specifically implicated in development and progression of several cancer types [140,150]. Our group described the involvement of an aberrant expression of metabotropic glutamate receptor 1 (mGluR1) in melanomagenesis [151].

### 4.2. mGluR1-Driven Melanoma

mGluR1 expression and activation were found to induce the transformation of normal melanocytes into melanoma cells. This was discovered through establishing transgenic mice where one out of five founder mice developed pigmented lesions. These lesions were subsequently identified histologically to be melanoma [152,153]. The transgenic mice were established to study cell differentiation by the use of a small genomic DNA fragment that showed the ability to induce adipocyte differentiation in vitro, as confirmed by biochemical and histological characterization [154,155]. Molecular analysis revealed that an insertional mutagenesis had occurred in the transgenic mouse (TG-3) that presented with pigmented lesions. The insertion of the small genomic DNA fragment, termed clone B, into the intron 3 region of the *Grm1* gene coding for mGluR1 was concurrent with the deletion of a 70 kB host DNA fragment, which led to the aberrant expression of mGluR1 in melanocytes. Normal melanocytes do not express mGluR1 [151]. To confirm the role of the ectopically expressed mGluR1 in the transformation, a second transgenic mouse line, named Tg(*Grm1*)EPv, was generated using the cloned murine *Grm1* cDNA expression that was driven by a melanocyte-specific promoter dopachrome tautomerase. These mice presented with pigmented lesions that were similar in onset and progression to the tumors of TG-3 mice, confirming the role of aberrant mGluR1 expression in the transformation of normal melanocytes into melanoma cells [151]. Analysis of human melanoma cell lines and biopsy samples revealed mGluR1 expression in 92% (no. = 25) of human melanoma cell lines, 65% (no. = 175) of primary and metastatic human melanoma biopsy samples and 33% of human dysplastic nevus samples. In contrast, in human melanocytes mGluR1 expression was not detected [151,156,157]. Further studies by several other groups demonstrated that functional mGluR1 was necessary for not only the development but also the progression of melanoma tumors in vitro and in vivo [157,158,159,160]. In mGluR1-positive tumors, levels of extracellular glutamate were shown to be elevated, suggesting the establishment of an autocrine/paracrine loop that promotes increased glutamatergic signaling through mGluR1 (Figure 2) [150].

The activation of mGluR1 initiates downstream signaling through the MAPK pathway. The PI3K/AKT cascade is thought to be activated by mGluR1 in an indirect manner via the activation of Src and the subsequent launch of insulin-like growth factor receptor 1 (IGF-R1) signaling cascade. Stimulation of these pathways provides signals for cell proliferation, resistance to cell death, angiogenesis, metastasis, dysregulation of cellular metabolism and immune evasion [157,158,159,161,162,163]. Another prominent feature of cancer cells, neoangiogenesis, was reported to be promoted by mGluR1 activity through the induction of vascular endothelial growth factor (VEGF) and interleukin-8 (IL-8) via the PI3K/AKT/mTOR/HIF-1 pathway (Figure 2) [164]. mGluR1 was also suspected to be involved in the Deubiquitinase cylindromatosis (CYLD)–NF-κB axis. CYLD is a tumor suppressor, and downregulation of CYLD leads to increased levels of cyclin D and N-cadherin that promote cell proliferation, migration and invasion (Figure 2) [165]. A connection between CYLD expression and the onset of melanoma was demonstrated with Tg(*Grm1*)EPv. The homozygous loss of CYLD resulted in quicker cancer development and progression as well as increased neoangiogenesis [166]. Another study demonstrated that NF-κB, a transcription factor known for its involvement in cancer development and progression, is regulated by CYLD [167,168]. Earlier studies from our lab showed constitutive activation of NF-κB in mGluR1-positive melanoma cells, further suggesting a link between mGluR1 and CYLD/NF-κB [169]. We hypothesize that mGluR1 stimulates NF-κB through downregulation of CYLD to promote tumor growth; however, this theory requires further investigation and validation.

## 5. Therapeutic Potential to Treat mGluR1-Driven Melanoma and Other Cancers with Riluzole

### 5.1. Use of Riluzole in Melanoma

The encouraging data suggesting the contribution of mGluR1 to the onset and progression of melanoma delivers a promising target to explore for therapeutic purposes. Our group has assessed a small-molecule drug named riluzole (Rilutek^®^) that has been approved by the FDA for treatment of Amyotrophic Lateral Sclerosis (ALS). One of its known functions is the inhibition of glutamate export to the extracellular space. This reduces the amount of the ligand glutamate that is available to activate glutamate receptors and glutamatergic signaling. Riluzole was shown to reduce levels of extracellular glutamate in mGluR1-positive melanoma cells, decrease cell proliferation and increase cell apoptosis in vitro and in vivo. Furthermore, mGluR1-positive melanoma cells were more sensitive to riluzole than were normal melanocytes or mGluR1-negative melanoma cells, supporting the notion of riluzole being a selective agent in targeted therapy [158,170]. These findings were first translated in a phase 0 clinical trial for patients with advanced melanoma (stage III/IV) that used riluzole at the dose approved for ALS patients. mGluR1 expression was not a requirement for participation in this clinical trial; however, it was detected in all participants in the trial. This trial revealed a significant reduction in MAPK and PI3K/AKT signaling, as well as a decrease in FDG-PET intensity in 34% of paired pre- and post-treatment patient samples [171]. Based on these observations, a phase II clinical trial using riluzole monotherapy was conducted in patients with unresectable stage III/IV melanoma. A stable disease span of 23–56 weeks was observed in 46% of the patients. Downregulation of MAPK and PI3K/AKT pathways was also seen in this phase II trial in 33% of paired pre- and post-treatment tumor samples [172]. As previously reported for ALS patients, this phase II trial also revealed high variability in the levels of bioavailable riluzole among patients [172]. This observed variability may in part account for the lack of response in some patients. It was shown in several studies that the difference in the metabolic clearance of riluzole is likely to be contributed to by the inconsistent expression of the cytochrome P450 isoform CYP1A2, a key metabolic enzyme for riluzole [173]. On account of such variability, the prodrug troriluzole was developed with the goal of giving more consistent effects in patients by bypassing liver metabolism and increasing riluzole bioavailability. Troriluzole is proven to have equal or higher efficacy than riluzole in preclinical studies (Patent Publication US20210228549) [174]. Based on the results of the monotherapy clinical trials, riluzole has the potential to be beneficial to some melanoma patients; however, the option of using it as part of a combinational therapy presents as a more promising one.

The exact molecular mechanism by which riluzole inhibits the export of glutamate has not been elucidated. Our group demonstrated earlier that only mGluR1-expressing cells were arrested at the G2/M phase at 24 h in the presence of riluzole with subsequent apoptosis, as indicated by the accumulation of cells in subG1 by 48 h post-treatment. mGluR1-negative cells did not show such alterations in cell cycle profiles [158]. A high proportion of cells in G2/M phase indicates the repair of DNA damage. This was confirmed by increased levels of ROS and γH2AX, a marker of DNA double-stranded breaks, in riluzole-treated mGluR1-positive melanoma cells. It was further supported by elevated γH2AX levels in post-treatment melanoma specimens from patients with stable disease who participated in the phase II single-agent riluzole trial [175]. We further determined that riluzole specifically produces double-stranded DNA breaks and induces nonhomologous end joining (NHEJ) repair pathways [176]. Based on these results, we propose that riluzole may act by mediating the export of glutamate via the xCT antiporter. xCT facilitates the export of one molecule of glutamate in exchange for the import of one molecule of cystine. Cystine is then intracellularly reduced to cysteine that participates in the synthesis of glutathione (GSH) [177]. The decrease in the export of glutamate in the presence of riluzole in mGluR1-positive cells leads to the reduction in the import of cystine, diminished cysteine levels, and less GSH being synthesized for the clearance of ROS. Increased ROS accumulation within the cell ultimately leads to DNA damage, enhanced mutational burden and cell death (Figure 2). We showed that ROS levels are elevated only in riluzole-treated mGluR1-expressing melanoma cells with decreased levels of GSH [175]. Additionally, cells with higher expression of xCT were shown to be more sensitive to riluzole treatment, which corroborates our working hypothesis [178].

This presumed mode of action for riluzole suggests that combining riluzole with an anti-PD-1 inhibitor would be beneficial. Riluzole will promote tumor mutational burden via increased DNA damage and enhanced neoantigen production and mobilize anti-tumor immune cells to the tumor site. In line with this concept, a phase II riluzole monotherapy clinical trial demonstrated that patients with stable disease had increased immune cell infiltration compared to patients with progressive disease [172]. Taken together, these results suggest that the combination of anti-PD-1, an agent that enhances the ability of cytotoxic immune cells to identify and eliminate cancer cells, with troriluzole, the prodrug of riluzole, will enhance the cytotoxic effects of infiltrating immune cells. This hypothesis was tested in our lab by conducting a preclinical study using traditional graft models (allograft and xenograft) as well as the melanoma-prone transgenic mouse models described above. In order to enable easier monitoring of tumor progression, we crossed a TG-3 mouse with a hairless SKH mouse and derived TGS mice. TGS mice exhibit similar onset and progression of melanoma to those described for TG-3 and Tg(*Grm1*)EPv mice, and they mimic human melanoma development and progression [174].

Many in vivo preclinical therapeutic studies are short-term (4–6 weeks) and do not accurately recapitulate patients’ circumstances. Our unique TGS mouse model provided an opportunity to assess the longitudinal treatment response (up to 15 months in the studies of troriluzole combined with anti-PD-1) without obvious toxicity [174,179]. A sex-dichotomy in treatment response was observed in which male mice benefited from a lower dose of troriluzole than did female mice. When we increased the troriluzole dosage, we detected a rescued response rate in some female mice. We also observed a decrease in response across treatment modalities in both sexes, suggesting the emergence of resistant cell populations and/or a reduction in immune cell infiltrates [179]. Results from these in vivo studies suggest that the TGS model is a responsive and tractable system for evaluating therapeutic regimens for melanoma in a long-term immunocompetent setting. Troriluzole and nivolumab were shown to be safe and tolerable in patients with advanced and refractory solid tumors in a phase I clinical trial [180]. A phase II trial using this combination for melanoma patients with brain metastasis was initiated (NCT04899921) but had to be terminated due to poor enrollment during the COVID pandemic. However, an ongoing clinical trial of troriluzole and anti-PD-1 treatment for glioma patients shows promising results for the use of this combination as a therapeutic strategy [181].

### 5.2. Use of Riluzole in Cancers Other Than Melanoma

In the last decade, riluzole has been tested in various other cancers with some success. Beneficial effects were seen in breast, pancreas, colon, liver, bone, brain, lung, prostate, thyroid, blood-related and nasopharynx tumors [182,183,184,185,186,187,188,189,190,191,192,193,194,195,196,197,198]. Interestingly, not all of these cancers express mGluR1 or other members of the glutamate receptor family. Despite this, some of these cancers respond to riluzole treatment [185]. Furthermore, results obtained from some studies, mostly in breast cancer, demonstrate that the therapeutic effects of riluzole could be disconnected from its role as an inhibitor of glutamatergic signaling [185,188].

As a single-agent therapy in pancreatic cancer, riluzole was used in both in vitro and in vivo studies, where it showed efficacy by inhibiting the Wnt-β-catenin/TCF-LEF pathway. This pathway is involved in the expression of glutamate transporter, glutamine synthetase and glucose transporter 2, which modulate apoptosis, cell migration and autophagy among other markers and pathways [184,186]. In prostate cancer, riluzole was reported to have anti-invasive, as well as anti-proliferative, activity [183,189,191]. One of the potential mechanisms underlying its mode of action in prostate cancer is the downregulation of androgen receptor expression through the endoplasmic reticulum stress pathway and selective autophagy [191]. Riluzole also inhibited the growth of both estrogen receptor-positive (ER+) and triple negative (TNBC) breast cancer cells [185]. In liver cancer, riluzole demonstrated anti-proliferative activity in vitro and in vivo via the same mechanism as in melanoma by inducing cell cycle arrest, apoptosis and modulating levels of GSH and ROS [190]. In bone cancer, riluzole showed treatment efficacy both in vitro and in vivo, especially when it was delivered via iron oxide nanocage [192,193,195]. In osteosarcomas, the presence of riluzole was shown to alter the phosphorylation status of AKT/P70 S6 kinase, ERK1/2 and JNK1/2 signaling cascades involved in cell proliferation and cell survival [195].

Given the prominent role of glutamate signaling in the CNS, riluzole has been extensively used in brain cancer therapy [194,196,197,198]. It inhibited tumor cell growth in tumor stem-like-cell-enriched cultures isolated from two human glioblastomas (GBM) through the suppression of glucose transporter 3 expression and inhibition of the p-AKT/HIF1α pathway. Additionally, downregulation of DNA (cytosine-5)-methyltransferase 1 (DNMT1), which is involved in the hypermethylation of various tumor-suppressor genes, and poor prognosis in GBM were reported [197]. In glioma U87 cells, the inclusion of riluzole in in vitro cultures and in vivo xenografts led to the suppression of mGluR1 signaling and significant decreases in tumor cell proliferation, potentially through reduced phosphorylation of PI3K, AKT, mTOR and P70S6K, and this has been confirmed in several studies [196]. In addition to its efficacy in brain tumors, riluzole’s therapeutic effectiveness has also been investigated in metastasis to the brain [199,200]. Combining riluzole with other agents, including radiation therapy, seems to create a promising approach to battle melanoma metastasis to the brain [199].

In general, combinational therapy has proven to be an advantageous strategy for many diseases in the last decade. Riluzole has also been tested in combination with other therapeutics to achieve improved synergistic outcomes. In liver cancer, riluzole was coupled with sorafenib, a protein kinase inhibitor, and the combination therapy showed stronger inhibition of cell proliferation than did either compound alone [190]. A phase I clinical trial using the combination of riluzole and sorafenib in patients with advanced solid tumors was completed. Stable disease was reached in 36% of patients, and partial response was observed in 2.9% of patients. This combination was also found to be safe and tolerable [201]. In TNBC, synergistic effects were observed with riluzole combined with paclitaxel, a chemotherapeutic agent, in vitro in cell lines that were resistant to paclitaxel and in vivo in a xenograft model [202]. In colorectal cancer, together with cisplatin, a conventional chemotherapeutic, riluzole proved to be effective in inhibiting growth of cisplatin-resistant cell lines [203]. In another study, Li et al. developed a platinum (IV)–riluzole conjugate compound that displayed IC_50_ values significantly exceeding those of cisplatin. It increased DNA damage and cell apoptosis and also suppressed invasion in the HCT-116 colon cancer cell line. In GBM, it was found that, together with an mTORC1/2 inhibitor PP242, riluzole synergistically inhibited the proliferation of GBM cell lines and a patient-derived cell line, as well as tumor growth in xenograft models. This synergistic result seems to derive from reduced levels of cyclin D1, c-MYC, G1 arrest and induced apoptosis [204]. In another study, riluzole was combined with temozolomide, a chemotherapeutic agent, and tested on GBM cell lines as well as in an orthotopic mouse allograft model of O6-methylguanine DNA methyltransferase (MGMT)-positive GBM. Riluzole was found to enhance the anti-tumor activities of temozolomide in MGMT-positive but not MGMT-negative GBM cell lines [205]. Furthermore, riluzole was also found to sensitize some tumor types to ionizing radiation [194,200,206]. These investigations of riluzole using in vitro and in vivo assessments further point to the promise of using riluzole or the prodrug troriluzole in improved therapeutic strategies for various cancer types [207].

However, despite the investigation of riluzole as a therapeutic option in numerous other cancers, the most comprehensive studies with riluzole or troriluzole were performed in melanoma. Our recently completed preclinical longitudinal studies in TGS mice treated with troriluzole and/or anti-PD-1 are the first preclinical studies to show a loss of response across treatment modalities, suggesting that intrinsic and/or acquired resistance must be involved in the reduced treatment efficacy as seems to be the case in many human patients [174,179]. This also suggests that the TGS model is suitable to use for not only tractable treatment responses but also for gradual resistance development over a prolonged period, which provides an opportunity to examine the early steps in the onset of treatment resistance. To date, little is known about the origins of resistance to riluzole in ALS or different cancer types. One possibility is that the acquired resistance to riluzole may be related to the upregulation of the multidrug resistance protein 1 (MDR1) reported in ALS mouse models and patient samples [208,209,210,211,212,213]. MDR1 is a cell membrane protein that serves as an efflux pump for many small-molecule compounds. Riluzole was shown to be a substrate for MDR1, supporting the notion that MDR1 is likely to be in part responsible for the resistance to riluzole [208,214]. Other evidence suggests that riluzole may be a substrate for breast cancer resistance protein (BCRP) at the blood–brain barrier (BBB) as well. Therefore, administration of riluzole together with MDR1/BCRP inhibitors may improve its therapeutic efficacy [215,216]. To our knowledge, there are currently no reports on the basis for resistance to riluzole in melanoma or other cancers.

## 6. Resistance to Different Treatment Options in Melanoma

Cancer therapies have advanced significantly in the last decade with increasing success in treatment responses. However, despite this improvement, the majority of therapeutic agents are effective only for a limited time period before the responses decrease and resistance develops. Treatment resistance has been postulated to derive from intrinsic and/or acquired genetic and/or functional changes from the status quo in cells. Intrinsic resistance comprises the inherent factors that allow tumors to tolerate treatment modalities from the beginning of treatment and continue to progress. Acquired resistance encompasses the adaptation of tumors to treatments via mutations, rewiring of signaling and/or metabolic pathways, epigenetic changes and modulation of the tumor/stromal microenvironment.

Tumors are heterogeneous entities that consist of cancer cells with different genetic profiles. Therefore, multiple cell populations exist within the same tumor. Both acquired and/or intrinsic resistance are likely to occur in one or more of such cell populations that will progress to form tumors that are non-responsive to the treatment (Figure 3). In this review, we discuss the current knowledge of the underlying mechanisms of resistance to targeted therapies and immunotherapy in melanoma. We do not claim this review to be an exhaustive source of information, but we are summarizing the most common and well-studied types of acquired resistance and strive to provide a general overview of the topic.

## 7. Resistance to Targeted Therapies

### 7.1. MAPK Proteins

The most prevalent mutation in melanoma patients is the BRAF V600E mutation. As such, resistance to BRAF-targeted therapy is one of the best studied topics in the therapeutic resistance of melanomas. For BRAF/MEK inhibitors, it is the specific reactivation of the MAPK pathway that is the most prominent mechanism of resistance [217,218,219,220,221] (Figure 4). The development of MEK inhibitors was initiated to overcome resistance to small molecules targeting mutated BRAF. RAS protein can activate several forms of RAF, including ARAF, BRAF and CRAF, which mediate phosphorylation of MEK, followed by phosphorylation of ERK and its downstream effectors that promote cell proliferation and survival signals. RAS activates RAF by relocating RAF to the plasma membrane and causing dimerization. BRAF inhibitors only bind to a mutated version of the protein, specifically BRAF V600E. Melanoma cells that harbor this mutation exhibit low levels of activated RAS that are less likely to mediate BRAF V600E dimerization and direct downstream signaling. However, increased RAS activity will lead to BRAF signaling even in the presence of BRAF inhibitor. The BRAF in the dimer that is not bound to the inhibitor will initiate downstream signaling desensitizing tumor cells to the targeted therapy (Figure 4) [222]. Another resistance mechanism involves a secondary mutation in BRAF that can alter its function and cause a loss of response to treatment. One such mutation was characterized by Poulikakos and colleagues: p61BRAF(V600E) protein, which enhanced dimerization in a RAS-independent manner [223]. Therefore, mutations in RAS or other upstream proteins that increase its signaling and drive RAF-dimer formation can contribute to the acquired resistance to targeted BRAF inhibitor treatments. Examples of such mutations are gain-of-function (GOF) mutations in RAS and loss-of-function (LOF) mutations in NF1 that have been described by several groups and demonstrated to drive resistance to BRAF inhibitors [219,220]. Some other possibilities include mutation-independent structural changes in BRAF itself that are mediated by alternative splicing, dimerization of BRAF and the constitutive activation of the MAPK pathway regardless of the status of upstream proteins [223]. The significance of BRAF dimerization in driving resistance was confirmed in a study using patient-derived xenograft models (PDX) established from metastasized human melanoma tumors. Duplications of the BRAF kinase domain led to resistance in approximately 10% of PDXs, which was verified in a patient cohort [224]. Some other means to drive BRAF-dimer formation include upregulated CRAF expression and increased copy numbers of BRAF [217].

Activation of the MAPK pathway downstream of BRAF has been noted as another way to induce resistance to BRAFi, including somatic mutations in MAP2K1 coding for MEK1 protein or indirect activation of MEK through MAP3K8, a MAPK substrate [225]. Further downstream in the MAPK signaling cascade, ERK has been implicated in acquired resistance to BRAFi through the loss of stromal antigen 2 or 3 (STAG2 or STAG3), which encodes subunits of the cohesin complex that is important for DNA replication. Reactivation of ERK signaling is mediated by deregulation of the expression of dual specificity phosphatase 6 (DUSP6), an inhibitor and regulator of ERK activity in melanoma, which is mediated by CCCTC-binding factor (CTCF) and by the loss of STAG2. LOF in dual specificity phosphatase 4 (DUSP4) may also be implicated in this process [226].

### 7.2. Receptor Tyrosine Kinases

The upregulation and activation of other receptor tyrosine kinases (RTK) are other ways for melanoma cells to circumvent the activities of BRAF/MEK inhibitors [221]. The intricate network of cellular signaling pathways is highly interconnected, which means that changes in the activity of pathways other than MAPK can also influence the MAPK pathway indirectly. Increases in RTK ligands and growth factors also contribute to alterations in cellular signaling, reducing innate drug sensitivity and driving acquired resistance [217]. Key RTKs that were noted to be implicated in driving acquired resistance to BRAFi are MET, insulin-like growth factor 1 receptor (IFG-1R), epidermal growth factor receptor (EGFR), platelet derived growth factor receptor ß (PDGFRß) and hepatocyte growth factor (HGF) [227,228].

### 7.3. MITF and Phenotypic Switching

Microphthalmia-associated transcription factor (MITF) plays a critical role in melanocyte lineage survival by regulating cell survival and anti-apoptotic gene expression; it has also been implicated in therapeutic resistance [229]. Depending on its level in the cell, MITF can have different functions: high levels of MITF contribute to differentiation, moderate levels contribute to proliferation, and low levels contribute to invasiveness [229]. With such multifaceted functions, overexpression of MITF has been reported to reduce the therapeutic effects of BRAFi, MEKi and their combination therapy by increasing cAMP signaling pathway activity [217,230,231,232,233,234,235]. Low MITF levels were shown to contribute to drug resistance by upregulation of the AXL receptor tyrosine kinase (AXL-RTK). Melanoma cell lines harboring BRAF or NRAS mutations frequently showed low MITF levels and high AXL-RTK levels. MITF and AXL-RTL levels have been used as a predictor of early resistance to multiple targeted therapy drugs [236,237]. Changes associated with MITF and AXL expression patterns in melanoma cells were proposed as means for cells to withstand therapeutic effects by transitioning from one differentiation state to another, which is often referred to as phenotypic switching [238]. For example, migrating melanoma cells were shown to exhibit low pigmentation and reduced proliferation rates; however, once they settled in a secondary site in the organism, they switched back to a more typical melanoma phenotype with increased pigmentation and proliferation [239]. Frequently, slow-cycling cells can withstand treatment and re-emerge as tumors at a later time as most therapies target fast dividing cells [240]. Such decreases in melanoma cell proliferation have been linked not only to MITF levels but also to regulation by histone demethylase-mediated chromatin remodeling factors. An example of these chromatin remodeling factors is Jumonji/ARID domain-containing protein 1B (JARID1B). These factors are regulated by hypoxia and cytokine signaling; their activity also leads to an increase in PI3K/AKT signaling [241,242]. Although several studies have proposed the involvement of MITF in phenotype switching, the precise role of MITF in all of these processes remains to be elucidated [236,243,244].

### 7.4. NF-κB Signaling and Tumor Microenvironment

Another study identified NF-κB signaling downstream of AXL as being responsible for incurred resistance to BRAFi [237]. Tirosh et al. proposed distinct tumor-microenvironmental patterns, including cell-to-cell interactions, for melanoma cells resistant to BRAFi with MITF^low^/AXL^high^ profiles [243]. Apart from the significance of the MITF/AXL ratio in the development of resistance to BRAFi, MITF^low^/JARID1B^high^ and MITF^high^/PGC1α^high^ (PPARG coactivator 1 alpha) ratios also have been linked to drug resistance in melanoma [245,246].

Interestingly, a transcriptome and methylome analysis study of matched paired pre- and post-treatment melanoma samples showed that DNA mutations were not recurrently present in samples resistant to the treatment. In contrast, transcriptome alterations were very common, with high abundances of Yes-associated protein 1 (YAP1) and MET and low levels of lymphoid enhancer-binding factor-1 (LEF1). Additionally, altered methylation mediated by MAPK inhibition was observed in key regulatory genes and the transcriptome in the resistant samples in a time-dependent manner during treatment [217,247]. Enhanced activation of the NF-κB pathway, together with increased numbers of monocytes and M2 macrophages, suggest that MAPK inhibition may lead to changes in the tumor microenvironment and contribute to therapy resistance together with alterations within the tumor itself [247].

### 7.5. Parallel Signaling Cascades

An alternative path for the onset of resistance is the activation of parallel signaling cascades, such as PI3K/AKT/mTOR, through changes in the expression of negative regulators of the pathway: PTEN and/or RB1 (Figure 4) [248]. Identifying such pathways will provide options for overcoming acquired resistance. This strategy was utilized in a study by Greger and co-workers where they demonstrated in vitro synergistic effects of combining dabrafenib with trametinib [249]. Subsequently, the FDA granted approval for the use of this combination in patients with unresectable or metastatic melanoma and later extended its use to other tumor types harboring the same mutations. Another study identified ligand-independent ephrin type-A receptor 2 (EPHA2) signaling as a result of BRAFi resistance was associated with increased invasion and metastasis in an AKT-dependent manner. Based on this observation, inhibition of EPHA2 signaling was proposed to be a promising strategy to prevent the onset of metastasis in BRAFi-treated patients [250].

### 7.6. Autophagy

Autophagy occupies an ambiguous position in cancer therapy as both a tumor-promoting and tumor-inhibiting component [251]. Autophagy can promote tumor resistance to BRAFi by enhancing ATP secretion. Excessive extracellular ATP contributes to tumor cell migration and invasion, possibly through the purinergic receptor P2RX7, which is a player in the autocrine–paracrine pathway that promotes an invasive tumor cell phenotype [252]. Another mechanism that may be involved in autophagy-mediated BRAFi resistance is the induction of endoplasmic reticulum (ER) stress and activation of the TAM (TYRO3, AXL, MER) receptor pathway [238,253]. Targeting pathways that can increase ER stress could be a viable anti-cancer strategy. Cerezo et al. developed a compound that induces ER stress by targeting BiP/GRP78/HSPA5 and triggering the concomitant induction of autophagy and apoptosis in melanoma cells. They also confirmed that this compound inhibits tumor growth in vivo in a xenograft model using both BRAF-sensitive and -resistant cells [254]. In brain cancer, targeting autophagy was shown to inhibit tumor growth and increase cell death in BRAFi-resistant cells, which has been confirmed in a patient population [255].

### 7.7. MicroRNAs

MicroRNAs (miRNAs) are involved in the transcriptional regulation of numerous genes. Specifically, they were found to mediate the protein expression of genes involved in signaling cascades linked to BRAFi-associated resistance [256]. For example, miR-204 and miR-211 were shown to confer BRAFi resistance in vitro [257]. miR-204 is regulated by STAT3 in amelanotic melanoma cells, where it targets AP-1 complex subunit sigma-2 (AP1S2) and inhibits motility. miR-211 is a transcriptional target of MITF in melanotic melanoma cells that targets ER degradation-enhancing alpha-mannosidase-like 1 (EDEM1) of tyrosinase through the ER-associated degradation (ERAD) pathway. miR-211 also promotes pigmentation, which was shown to be an adaptive response to BRAFi treatment and to contribute to therapy resistance [258].

Results from these studies clearly point out the complexity associated with resistance to BRAF/MEK inhibitors. It is not surprising that no compounds have been developed and incorporated into existing therapeutic regimens to increase the effectiveness of therapy upon a loss of response to BRAFi or MEKi. Therefore, much work remains to be completed in this area.

## 8. Resistance to Immunotherapies

### 8.1. Tumor Mutational Burden

The rationale and principle of immunotherapy are based on sensitization of the host’s immune system. Therefore, reducing immune system responses to cancer cells is a logical approach for tumors to use to evade destruction. One way to do that is to diminish T-cell activity. Depending on the subtype, T-cells can have either immune-stimulating or immune-suppressive activity. Effector T-cells include several T-cell subtypes that respond to stimuli and mediate an appropriate immune response. T-cells become activated by interacting with a major histocompatibility complex (MHC) coupled with an antigen to initiate a T-cell immune response. Interruptions and errors in antigen presentation can lead to reduced immune responses; as such, antigen presentation is a process commonly hijacked by tumors to evade immune recognition. Antigens that are presented for T-cell recognition can be in different forms; frequently, mutated or broken-down proteins can serve as antigens in cancer cells. The number of such newly generated antigens in tumor cells is commonly described as the tumor mutational burden (TMB). Higher TMB usually denotes a larger number of mutations that increase antigenicity and promote immune system recognition of tumor cells. In general, melanoma patients with higher TMB showed better responses to immunotherapy and improved OS [259,260,261,262,263]. Evidence exists that TMB can even be used to predict therapy outcomes and the onset of immunotherapy resistance [264]. Immunotherapeutic agents naturally target cells with higher TMB and establish selective pressure for the survival of cancer cells with lower TMB that do not respond well to therapy but will eventually become the source of a resistant tumor [265]. Using therapeutic approaches that target ‘cold’ unresponsive tumors together with immunotherapy will help to overcome this hurdle.

### 8.2. Antigen Presentation: LAG-3 and B2M

Another way in which cancer cells modulate antigen presentation is through the upregulation of LAG-3, an immune checkpoint receptor often expressed on activated immune cells [266]. LAG-3 has a dual role, being both an activator of T-cells when bound to MHC-II in the process of antigen presentation, and a T-cell suppressor when it is located on T-cells. Because of this ambiguity, LAG-3 has been used as a target for therapeutic intervention and a marker to predict poor therapy responses [267,268,269]. Elevated levels of soluble LAG-3 in serum from patients receiving anti-PD-1 therapy have been linked to poorer responses. Increased tumor infiltration of LAG-3-positive T-cells, T-cell immunoglobulin and mucin domain-containing protein 3 (TIM3) were also associated with shorter PFS. TIM3 is a co-inhibitory receptor expressed on IFN-γ-producing T-cells involved in immune tolerance [268,270]. As such, patients who participated in a combination clinical trial using PD-1 antibody, nivolumab, and a monoclonal antibody targeting LAG-3, namely relatlimab, showed improved PFS [39]. Beta-2-microglobulin (B2M) is another protein involved in antigen presentation that was implicated in the onset of resistance to immunotherapy in melanoma patients. In a paired-patient-sample study, a truncating mutation in the gene coding for B2M was identified as being responsible for the loss of MHC-I on the cell surface [271].

### 8.3. IFN Signaling and Other Cellular Regulators

Once T-cells are activated, IFN-γ serves to amplify immune responses by recruiting leukocytes, macrophages and natural killer T-cells. Over time, IFN-γ induces the expression of several factors, including indolamine-2,3-dioxygenase (IDO), to reduce this immune response [272,273]. Additionally, IFN-γ has been shown to induce PD-L1 expression and limit anti-PD-1-related immune responses [274,275]. PD-L1 upregulation is a major mechanism involved in acquired resistance to immunotherapy. Several other regulators, including PTEN, nucleophosmin (NPM)/anaplastic lymphoma kinase (ALK), STAT3, EGFR and nuclear factor erythroid 2-related factor 2 (NRF2), were also reported to be involved in this process [276,277,278,279,280]. Carcinoembryonic antigen cell adhesion molecule-1 (CEACAM1) is another target in IFN-γ signaling that heterodimerizes with TIM-3 and promotes T-cell exhaustion to suppress the immune system [281]. Both IFN-α and IFN-γ receptors regulate Janus kinase 1 (JAK1) and/or Janus kinase 2 (JAK2) activity that have been implicated in the development of resistance to anti-PD-1 therapy in several studies, including one using paired patient samples [271,282]. LOF mutations in JAK1 and JAK2 that were concurrent with the deletion of their wild-type alleles were found to be responsible for the onset of resistance in several patients [271].

Stimulator of interferon genes (STING) is an ER protein that is responsible for the regulation of IFN and chemokine signaling [283]. STING expression promotes an increase in MHC-I expression, antigenicity and lysis by tumor-infiltrating T-cells (TILs) via the upregulation of type I IFN signaling and the activity of the CXCL10 chemokine [284]. In one study, the induction of STING signaling improved anti-PD-1 and anti-CTLA-4 therapeutic responses by increasing IFN signaling [285]. However, defects in STING signaling promote tumor progression by protecting melanoma cells from increased immune recognition by TILs [284]. Based on these results, several studies reported on the development of drugs that induce STING signaling in melanoma cells [286,287].

High expression of sphingosine kinase 1 (SK1), a protein that catalyzes phosphorylation of sphingosine to give sphingosine-1-phosphate, is reported to be associated with immunotherapy resistance in melanoma patients. SK1 was proposed to regulate the process of lymphocyte trafficking and differentiation. The exact mechanism of resistance involving SK1 has not been established. However, its inhibition enhanced the efficacy of anti-CTLA-4 and anti-PD-1 therapy in vitro [288,289].

Another protein implicated in immunotherapy resistance is nucleotide-binding domain, leucine-rich containing family, pyrin domain-containing-3 (NLRP3). It induces an inflammasome signaling cascade in response to anti-PD-1 therapy, resulting in the recruitment of granulocytic myeloid-derived suppressor cells (MDSCs), immune cells known for their ability to suppress immune response, to the tumor site and the decreased efficacy of anti-PD-1 therapy [290].

### 8.4. Regulatory T-Cells

Regulatory T-cells are another type of immunosuppressive cell whose main function is to decrease proliferation and downregulate the activity of effector T-cells in order to maintain appropriate immune responses and avoid excessive immune reactions or autoimmunity. Within resistance to immunotherapy in melanoma, levels of regulatory T-cells expressing forkhead box P3 (FOXP3) are elevated. Their mobilization is dependent on the presence of CD8^+^ T-cells and is driven by the production of CCR4-binding chemokines and increased proliferation [274].

### 8.5. Phenotypic Switching

Similar to acquired resistance to targeted therapy, phenotypic switching occurs in cancer cells as a mechanism of resistance to immunotherapy. It involves tumor necrosis factor α (TNFα) signaling in tumor-specific T-cells during the T-cell-mediated immune response to cancer cells. TNFα is a cytokine responsible for modulating inflammatory responses. Its signaling has been associated with the reduced expression of genes that are specific to melanoma cells and an increase in the expression of genes that trigger de-differentiation, resulting in a profile more similar to neural crest-derived cells/immature melanocytes [291].

### 8.6. Epigenetic Regulation

Finally, epigenetic change is another way to influence cellular proteomic profiles, via processes of the methylation, demethylation and acetylation of DNA or histone proteins. In immunotherapy resistance, epigenetic changes can influence the expression of proteins that promote resistance [292]. DNA hypermethylation reduces expression of 4-1BB (CD137), which helps to stimulate T-cell and B-cell proliferation as well as dendritic cell maturation. Upregulated IL-10 and PD-L1 expression mediated through histone deacetylase 6 (HDAC6) decreases immune response to tumors, as discussed above. Mutations in enzymes that regulate epigenetic functions also influence immune responses to tumors. An example involves mutations in the histone methyltransferase EZH2, whose activity causes reduced expression of RAS association domain-containing protein 5 (RASSF5) and integrin beta chain-2 (ITGB2); these are proteins that are involved in anti-tumor response [293]. KDM5B, another histone demethylase, has also been implicated in the onset of resistance to immunotherapy by recruiting SETDB1, an H3K9 methyltransferase whose overexpression has been linked to reduced anti-tumor activities in several cancers [293,294,295,296]. However, to our knowledge there are no reports regarding their role(s) in resistance to immunotherapy in melanoma.

The loss of response to immunotherapy treatment has been extensively studied, and this review does not claim to be an exhaustive source on the topic. For a more detailed review on the mechanisms of immune evasion in melanomas, please refer to Eddy et al., 2020 [84].

## 9. Conclusions and Future Directions

Significant progress has been made in melanoma therapy in the past two decades with the development of targeted therapeutics against mutated BRAF/MEK and the introduction of immunotherapy. BRAF/MEK inhibitors significantly improved patient responses and outcomes in comparison to previously available options of chemotherapy and radiation therapy. However, their efficacy and ability to overcome resistance remains to be improved. Based on the outcomes from clinical trials conducted in the last ten years, immunotherapy seems to achieve better efficacy for overall patient survival. But despite the advantages that immune checkpoint inhibitors offer, their efficacy as single agents remains at approximately 50%, and this is often accompanied by AEs [34,39,73]. Using combinations of different therapeutical agents (combining different molecular-tumor-targeted inhibitors with immune checkpoint inhibitors against different immune-related receptors) has been efficacious for improving patient responses. However, such increased effectiveness often leads to more severe AEs [34]. Therefore, further exploration of new targets and development of new therapeutic agents are still needed to maximize the benefits of treatment and minimize side effects.

An additional challenge for cancer therapy is the onset of resistance to treatment for nearly every type of drug. Investigations are ongoing to understand the underlying mechanisms in intrinsic and acquired resistance to therapeutics. Many of them are associated with the regulation of genes in redundant pathways and/or epigenetic regulation [217,238,292]. Understanding the loss of response to therapy at the molecular level will allow the design of novel therapeutic options based on rational strategies. Apart from that, the current paradigm for preclinical investigation of potential drug candidates—the ‘in vitro to in vivo’ model—is limited; the use of 3D culture approaches and suitable in vivo experimental models are expected to contribute significantly to assessment and the better prediction of long-term outcomes for many candidate therapeutics.

## Figures and Tables

**Figure 1 cancers-16-01571-f001:**
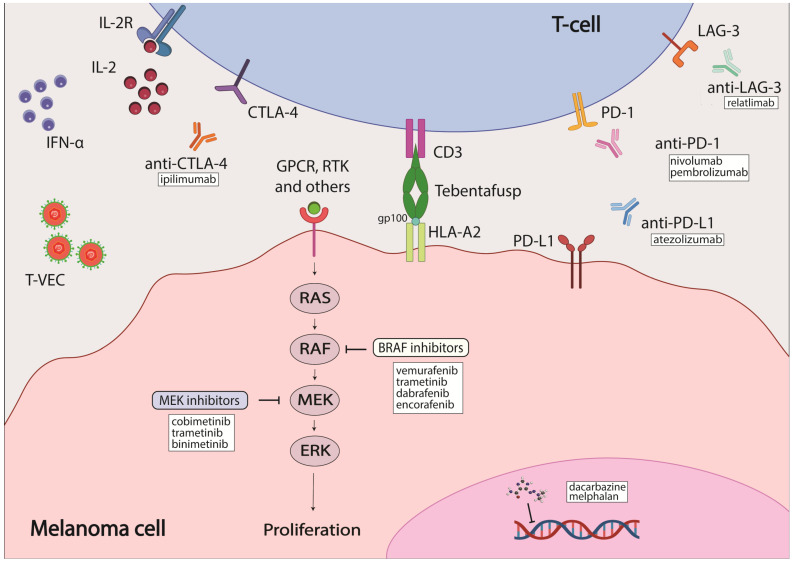
Melanoma therapies currently approved by the FDA. Some elements of the figure were created with BioRender.

**Figure 2 cancers-16-01571-f002:**
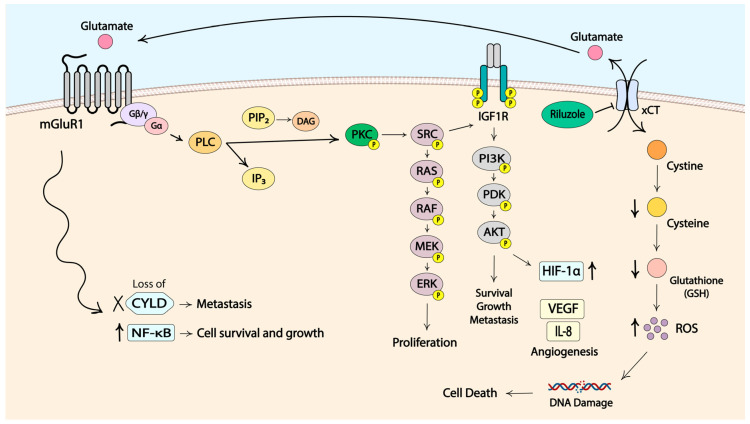
Molecular signaling pathways in mGluR1-positive melanoma cells. Some elements of the figure were created with BioRender.

**Figure 3 cancers-16-01571-f003:**
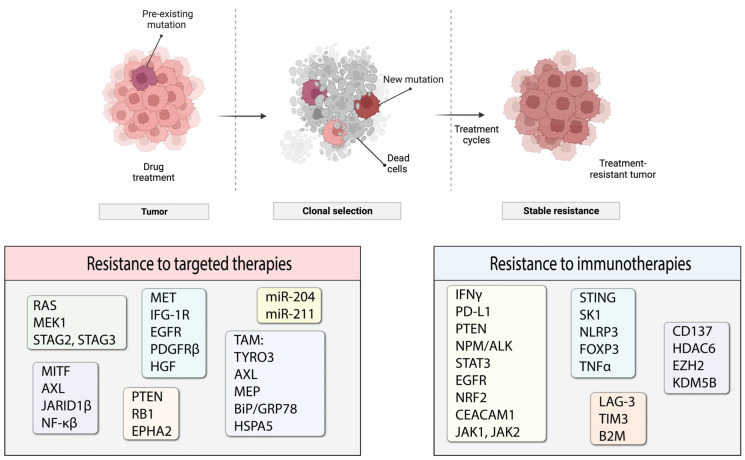
Process of resistance development and molecules likely to participate in the emergence of resistance. Some elements of the figure were created with BioRender.

**Figure 4 cancers-16-01571-f004:**
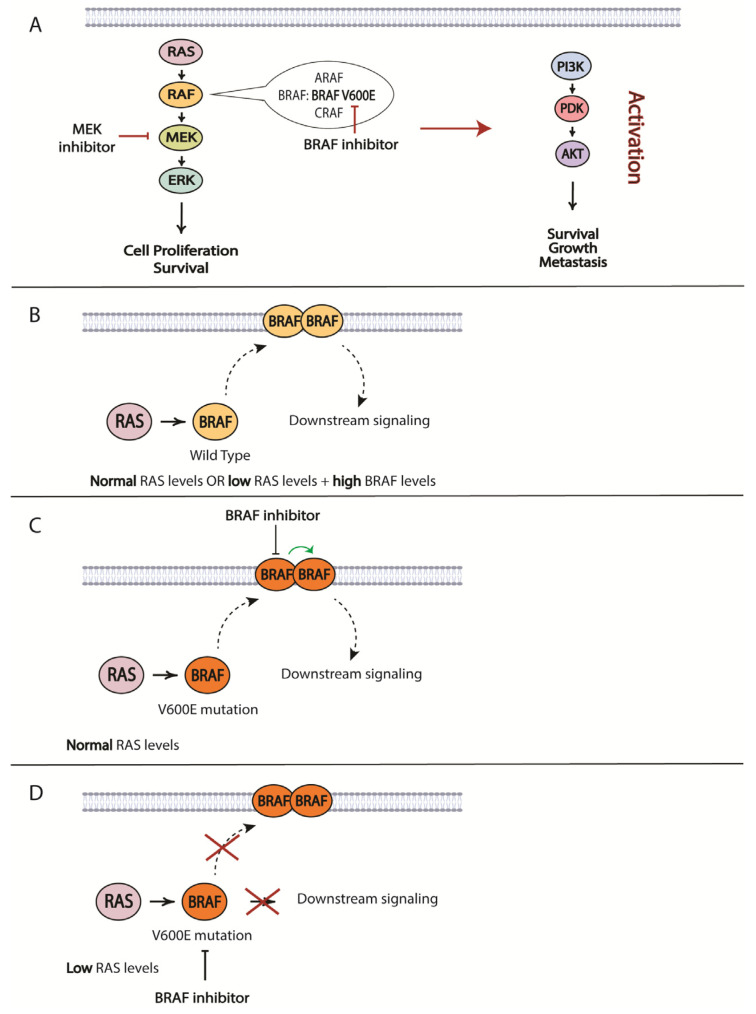
Some of the possible mechanisms of resistance to BRAF and MEK inhibitors. (**A**) Silencing of MAPK signaling pathway using BRAF/MEK inhibitors can lead to the activation of PI3K/AKT signaling pathway, resulting in circumvention of the inhibitors’ effect. (**B**–**D**) Mechanisms of BRAF signaling activation in the presence of a BRAF inhibitor and different levels of RAS. Some elements of the figure were created with BioRender.

**Table 1 cancers-16-01571-t001:** Therapeutics currently approved by the FDA for treatment of patients with melanoma tumors [18].

Name	Therapy Type	Disease Conditions	Year of Approval	Details	Limitations	References
Dacarbazine	Chemotherapy	Metastatic disease	1975	DNA-alkylating agent	Low efficacy Severe AEs	[16,19]
Recombinant Interferon Alfa-2b	Immunotherapy	Previously operated-on melanoma Used as adjuvant therapy For patients with high risk of recurrence	1995	Recombinant form of the protein interferon alpha-2	Low efficacy	[20,21]
Aldesleukin	Immunotherapy	Metastatic disease	1998	Human recombinant interleukin-2	Low efficacy	[22,23]
Vemurafenib	Targeted therapy	Unresectable or metastatic melanoma with a BRAF mutation	2011	BRAF inhibitor	Medium efficacy Acquired resistance is frequent	[24,25,26]
Ipilimumab	Immunotherapy	Unresectable or metastatic melanoma Used as preventative measure against recurrence	2011	Antibody against CTLA-4	Medium efficacy	[27,28]
Nivolumab	Immunotherapy	Unresectable or metastatic melanoma Used as adjuvant therapy	2014	Antibody against PD-1	Medium efficacy	[29,30]
Pembrolizumab	Immunotherapy	Unresectable or metastatic melanoma Used as preventative measure against recurrence	2014	Antibody against PD-1	Medium efficacy	[31]
Cobimetinib + vemurafenib	Targeted therapy	Unresectable or metastatic melanoma with a BRAF V600E or V600K mutation	2015	MEK inhibitor + BRAF inhibitor	High rate of AEs	[32]
Talimogene Laherparepvec (T-VEC)	Immunotherapy	Unresectable and metastatic melanoma Used as a local treatment in recurrent melanoma	2015	Oncolytic virus	Medium efficacy	[33]
Ipilimumab + nivolumab	Immunotherapy	Unresectable or metastatic melanoma	2015	Antibody against CTLA-4 + antibody against PD-1	High rate of AEs	[34]
Binimetinib + Encorafenib	Targeted therapy	Unresectable or metastatic melanoma	2018	MEK inhibitor + BRAF inhibitor	High rate of AEs	[23]
Dabrafenib + trametinib	Targeted therapy	Previously operated-on melanoma that spread to lymph nodes with BRAF V600E or V600K mutations Unresectable or metastatic melanoma	2018	BRAF inhibitor + MEK inhibitor	High rate of AEs	[35,36]
Atezolizumab + cobimetinib + vemurafenib	Immunotherapy + targeted therapy	Metastatic unresectable melanoma with BRAF V600E mutation	2020	Antibody against PD-L1 + MEK inhibitor + BRAF inhibitor	High rate of AEs	[37]
Tebentafusp-tebn	Immunotherapy	Unresectable or metastatic uveal melanoma	2022	T-cell receptor-bispecific molecule that targets both glycoprotein 100 and CD3	Not enough data available yet	[38]
Nivolumab + relatlimab	Immunotherapy	Metastatic or unresectable melanoma	2022	Antibody against PD-1 + antibody against LAG-3	High rate of AEs	[39,40]
Melphalan	Chemotherapy	Unresectable uveal melanoma with hepatic metastases	2023	DNA-alkylating agent	Medium efficacy	[41]
Lifileucel	Cellular therapy	Unresectable or metastatic melanoma that previously had been treated with anti-PD-1 and/or BRAFi	2024	Tumor-derived autologous T-cell immunotherapy	Not enough data yet—phase III of clinical trial is ongoing	[42]

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
