# Peer review of "Current State of Melanoma Therapy and Next Steps: Battling Therapeutic Resistance"

_cancers, 2024, doi:10.3390/cancers16081571_

Round 1

Reviewer 1 Report

Comments and Suggestions for Authors

Dear authors

I suggest major revision and editing errors.

Line 85 "BRAF" not "CRAF"

I would ask you to only describe uveal melanoma as eye cancer. I am not agree to define "subtype of UM" but subtypes of eye (ocular) cancer. And I would ask you to explain, in the chapter about UM, that we have more studies on uveal melanoma but in your review you cite a little part of knowledge about this rare disease.

Greetings

Thank you

Author Response

We would like to thank all the reviewers for their scholarly reviews and valuable suggestions. We have modified the manuscript and incorporated recommendations in this revised version. Below are our point-by-point replies to the concerns raised by each reviewer.

Reviewer 1

Line 85 "BRAF" not "CRAF".

Sorafenib is an inhibitor that targets both BRAF and CRAF. Therefore, the use of CRAF was intentional in this part of manuscript. Please refer to the following publications about sorafenib’s action on both BRAF and CRAF, where CRAF is often referred to as Raf-1 and sorafenib – as BAY 43-9006:

  • https://pubmed.ncbi.nlm.nih.gov/15466206/
  • https://pubmed.ncbi.nlm.nih.gov/12369853/
  • https://www.ncbi.nlm.nih.gov/pmc/articles/PMC2898184/
  • https://pubmed.ncbi.nlm.nih.gov/15035987/

I would ask you to only describe uveal melanoma as eye cancer. I am not agree to define "subtype of UM" but subtypes of eye (ocular) cancer. And I would ask you to explain, in the chapter about UM, that we have more studies on uveal melanoma but in your review you cite a little part of knowledge about this rare disease.

This has been revised.

Reviewer 2 Report

Comments and Suggestions for Authors

The manuscript idea is very nice

I like the review idea. However, it is not well-presented by the authors.

I would recommend the construction of more figures shwing the mechanisms of resistance and the new therapeutic apporaches to halt the resistance to conventional treatments for melanoma patients

Also, i would recommened the addition of a special section on the potential of PDT in the treatment of melanoma and its integeration with nanoformulations and phytochemical products because it has recently been evolved as a magical multi-factorial treatment apprach for melanoma patients For example: https://pubmed.ncbi.nlm.nih.gov/37689125/

Also Tables showing the array clinical trials that is currently ongoing to provide novel therapeutic approached for the melanoma patients should be included

Comments on the Quality of English Language

Minor revisions is required concerning the English and linguistic mistakes

Author Response

We would like to thank all the reviewers for their scholarly reviews and valuable suggestions. We have modified the manuscript and incorporated recommendations in this revised version. Below are our point-by-point replies to the concerns raised by each reviewer.

Reviewer 2

I would recommend the construction of more figures showing the mechanisms of resistance and the new therapeutic approaches to halt the resistance to conventional treatments for melanoma patients.

Thank you for this wonderful idea; however, to create figures on different mechanisms of resistance is a bit difficult, because most of these potential mechanisms are not well studied and, therefore, lack details to describe their mechanism of action. Rather, there are a few details known about their potential involvement in therapeutic resistance. We added a figure on several common mechanisms of resistance to BRAF, which are some of the better studied mechanisms (figure 4), and a figure showing conventional and the latest therapeutic options to treat melanoma (figure 1).

Also, I would recommended the addition of a special section on the potential of PDT in the treatment of melanoma and its integration with nanoformulations and phytochemical products because it has recently been evolved as a magical multi-factorial treatment apprach for melanoma patients For example: https://pubmed.ncbi.nlm.nih.gov/37689125/

A section on the potential of PDT is now added to the manuscript.

Also Tables showing the array clinical trials that is currently ongoing to provide novel therapeutic approached for the melanoma patients should be included.

Currently, there are over 270 active not recruiting ongoing clinical trials for melanoma patients. Many of them are new combinations of already existing drugs. The goal of this review is to summarize the knowledge on already existing therapeutics to treat melanoma and possible mechanisms of resistance to these therapeutics. Providing information on the drug candidates currently in trial is certainly informative, but beyond the scope of the current manuscript.

Reviewer 3 Report

Comments and Suggestions for Authors

In the current review article authors have compiled in depth and discussed therapeutic options for the treatment of melanoma. In particular, targeted therapies and immune therapies were focused with detailed mechanistic discussion on therapeutic resistance. Authors have presented a well drafted review by collecting information from highly scientific and reputed literature, while their method of selection was greatly unique. Overall, the efforts in the current review are enormous. There are only a few suggestions to improve the final quality of the review after which it can be accepted.

1.     Authors need to focus on presenting an individual table for the approved targeted and immuno therapies with their outcome, limitation, and specific details if any. This will represent overall summary and shall be helpful for readers.  

Author Response

We would like to thank all the reviewers for their scholarly reviews and valuable suggestions. We have modified the manuscript and incorporated recommendations in this revised version. Below are our point-by-point replies to the concerns raised by each reviewer.

Reviewer 3

In the current review article authors have compiled in depth and discussed therapeutic options for the treatment of melanoma. In particular, targeted therapies and immune therapies were focused with detailed mechanistic discussion on therapeutic resistance. Authors have presented a well drafted review by collecting information from highly scientific and reputed literature, while their method of selection was greatly unique. Overall, the efforts in the current review are enormous. There are only a few suggestions to improve the final quality of the review after which it can be accepted.

Authors need to focus on presenting an individual table for the approved targeted and immuno therapies with their outcome, limitation, and specific details if any. This will represent overall summary and shall be helpful for readers.

We have tried our best to highlight the outcomes and limitations of the therapeutics and included as much details as possible in the table added to the manuscript.

Reviewer 4 Report

Comments and Suggestions for Authors

This review article by Fateeva et al summarizes the currently available therapeutic options for melanoma treatment and focus on the possible mechanisms that drive therapeutic resistance.

The review is of interest but it can not be published in its current format. 

Major comments:

In fact, the article lacks coherence, is not well-structured, and is poorly written. English language is poor with too many mistakes. The article should be proofread by a native English speaker. Beside the corrections and comments I suggested above, I see that the title of the article is misleading considering the article content.

The introduction (from line 20 to line 402) included basic naïve information which distract the readers’ attention from the main message of this review article. I suggest either to omit completely this part or to summarize it in to 2 pages while conserving data relevant for the rest of the article.

Concerning the second part of the article (line 403 to line 855), which is the principal message of this review, it just need linguistic correction. It summarizes the work done in the laboratory of the authors. Besides, it correlates their findings with other researchers. This part is informative.

Concerning the last part (Line 856 to the end), the information concerning cell culture techniques is totally out of the spirit of this review article. The authors seem to be confused and they are confusing the reader, no fluidity in information provided. Too many information not relevant to this article. This part can be omitted, reduced to a single paragraph, or included in a separate review article.

Minor comments:

Line 216-222: change as following

“In 2024, the first cell therapy approach was approved by the FDA for the treatment of melanoma patients with unresectable or metastatic melanoma previously treated with anti-PD-1 and BRAFi [75]. Lifileucel is a tumor-derived autologous T-cell immunotherapy involving in vitro T-cells expansion.  This line of treatment demonstrated an OR rates of 31.5% in a phase II clinical trial [76]. A phase III clinical trial is currently underway with results expected in 2028-2030 (NCT05727904)”.

Line 228: remove “ONLY”

Line 231-232: Remove all this sentence or, if relevant, re-write it in an understandable way.

“For most therapeutic options available for AM and MM patients, retrospective studies and subgroup analysis of the general clinical trials had to be conducted to obtain data for these patients”.

Line 250: remove “as”

Line 258: “KIT has ambiguous functions” this piece of information is highly relevant to this paper, you need to expand it.

Line 273: Interleukin what??? Specify

Line 283-284: “confirming other analyses” à Mind English language

Line 293: response rate and not rates

Line 308: “of the last two decades” à in the past two decades

Line361: what is clone B?

Line 387: “Another prominent feature of cancer cells – angiogenesis” à It is neo angiogenesis

Line 495: “not all these cancers have the expression of mGluR1 or other members of the glutamate receptor family, and despite that, some of the tumors respond to riluzole treatment”. à May you give a possible explanation or reason on this effect? How riluzole treatment has a beneficial effects if cancers do not express mGluR1?

Line 552: “and an orthotopic” à in an orthotopic

Lacks transition words between paragraphs 

Comments on the Quality of English Language

English language is poor with too many mistakes. The article should be proofread by a native English speaker

Author Response

We would like to thank all the reviewers for their scholarly reviews and valuable suggestions. We have modified the manuscript and incorporated recommendations in this revised version. Please find our point-by-point replies to the concerns in the file attached.

Round 2

Reviewer 1 Report

Comments and Suggestions for Authors

dear authors ,

I think  your revision as asked is good. 

thank you 

Reviewer 2 Report

Comments and Suggestions for Authors

The revised manuscript is greatly improved and I found it fitting the scope of the journal and quality of manuscripts published in it 

Comments on the Quality of English Language

Minor revisions in proof has to be considered

Reviewer 4 Report

Comments and Suggestions for Authors

The paper is acceptable in the present form